# Sildenafil–Resorcinol Cocrystal: XRPD Structure and DFT Calculations

**Rafael Barbas [1], Vineet Kumar [2], Oriol Vallcorba [3], Rafel Prohens [1,2,* and Antonio Frontera [4,***

[1] Unitat de Polimorfisme i Calorimetria, Centres Científics i Tecnològics, Universitat de Barcelona, Baldiri Reixac 10, 08028 Barcelona, Spain; rafa@ccit.ub.edu

[2] Center for Intelligent Research in Crystal Engineering S.L., Parc Científic de Barcelona, Baldiri Reixac, 4-8, 08028 Barcelona, Spain; vkumar@circecrystal.com

[3] ALBA Synchrotron Light Source, 08220 Cerdanyola del Vallès, 08028 Barcelona, Spain; ovallcorba@cells.es

[4] Departament de Química, Universitat de les Illes Balears, Crta. de Valldemossa, km 7.5, 07122 Palma (Baleares), Spain

* Correspondence: rafel@ccit.ub.edu (R.P.); toni.frontera@uib.es (A.F.)

**Abstract:** Herein, the X-ray powder diffraction (XRPD) crystal structure of a new Sildenafil cocrystal is reported, where resorcinol has been used as the coformer. The crystal structure has been solved by means of direct space methods used in combination with density functional theory (DFT) calculations. In the structure, the Sildenafil and resorcinol molecules form cooperative hydrogen bond (HB) and π-stacking interactions that have been analyzed using DFT calculations, the molecular electrostatic potential (MEP) surface, and noncovalent interaction plot (NCI plot). The formation of O–H···N H-bonds between resorcinol and Sildenafil increases the dipole moment and enhances the antiparallel π-stacking interaction.

**Keywords:** sildenafil; X-ray powder diffraction; cocrystal; DFT analysis

## 1. Introduction

Sildenafil is a drug used in male patients with erectile dysfunction that acts by selectively inhibiting the cyclic guanosine monophosphate phosphodiesterase type 5 (Figure 1) [1,2]. The X-ray structures of Sildenafil and its citrate monohydrate salt have been previously reported [3,4]. Moreover, the solid-state structure of many salts and cocrystals is also available [5,6]. In addition, the cocrystallization of Sildenafil with acetylsalicylic acid [7] and also in the form of salicylate salt [8] has been used to mix a drug able to prevent strokes and myocardial infarctions with Sildenafil, which is not advised for patients with heart diseases. Moreover, we have previously discovered and analyzed, through a combined virtual/experimental cocrystal screen, new hybrid salt–cocrystal forms of Sildenafil [9].

**Figure 1.** Structure of Sildenafil.

Significant research has also been performed to synthesize and analyze a variety of Sildenafil solvates [5,10–12]. From a pharmaceutical point of view, these studies are very relevant because the presence of solvent molecules in the solid state may modify the solubility and the bioavailability of the drug [13–17]. Moreover, for drugs that are administered as solids like Sildenafil, the performance of the final product can change by the presence of solvates, which may have significant commercial interest in the pharmaceutical industry [15].

Herein, the X-ray powder diffraction (XRPD) crystal structure of a multicomponent solid form composed by Sildenafil free base and resorcinol in a (1:2) stoichiometry is reported. The cocrystal was previously discovered by some of us [9], although the crystal structure remained elusive until now. We focused this study on the analysis of the intermolecular H-bond between the resorcinol and Sildenafil molecules and its influence on the antiparallel stacking interaction between the pyrazolo[3,4-d]pyrimidine rings. For this purpose, this manuscript reports a complete theoretical study that consists of DFT calculations, molecular electrostatic potential maps, quantum theory of "atoms-in-molecules" (QTAIM) [18], and the noncovalent interaction plot (NCI plot) [19] analyses.

## 2. Materials, Experimental, and Theoretical Methods

### 2.1. Synthesis

Sildenafil (163 mg, 0.343 mmol, purchased from Polpharma, Warszawa, Poland) and resorcinol (76 mg, 0.690 mmol, purchased from Sigma-Aldrich, Darmstadt, Germany) were mixed and stirred in xylene (0.8 mL) during 4 days at 25 °C. The resulting suspension was filtered and dried under vacuum.

### 2.2. XRPD Analysis

XRPD pattern of the Sildenafil-resorcinol cocrystal was obtained on a PANalytical X'Pert PRO MPD diffractometer (see Supplementary Materials for details). The XRPD was indexed to a monoclinic cell of approximately 3530 $Å^3$ using Dicvol04, [20] (Figures of Merit: M = 72, F = 236), with the number of impurities equal to zero. The space group was assigned to $P2_1/a$ by using the systematic absences. The crystal structure was solved by the direct space strategy implemented in TALP [21], showing in the asymmetric unit one independent molecule of Sildenafil and two independent molecules of resorcinol (Z = 4). Previously reported single-crystal structures of both Sildenafil and resorcinol molecules have been used to set the soft molecular restraints for their geometry. The Rietveld method was used for the final refinement of the crystal structure. The FullProf in combination with DFT calculations (see ESI for further details, Figures S1–S3) was used in order to improve through geometry optimization the planarity of the aromatic rings and to locate the hydrogen atomic coordinates (Figure 2 depicts the final Rietveld plot). Relevant crystal data of refinement parameters are given in Table 1.

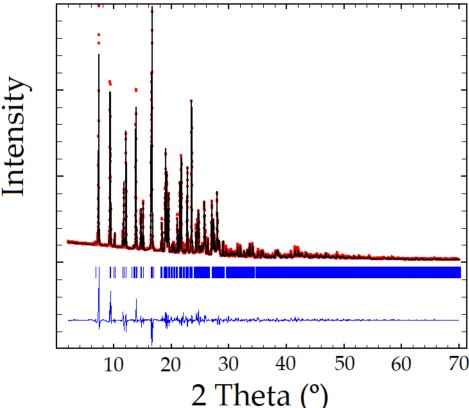

**Figure 2.** Rietveld plot of the Sildenafil–resorcinol cocrystal, $R_{wp}$ = 8.61%, $Chi^2$ = 107. Experimental powder XRD profile in red, the calculated one in black, and the difference in blue. Peak positions (in blue) are indicated using tick marks.

**Table 1.** Crystal data and structure refinement parameters for Sildenafil resorcinol cocrystal.

| Crystal Structure Data | |
|---|---|
| Empirical formula | $C_{34}H_{42}N_6O_8S$ |
| Formula Weight | 694.80 |
| Temperature (K) | 298(2) |
| Wavelength (Å) | 1.5406 |
| Crystal system | Monoclinic |
| space group | $P\,2_1/a$ |
| | 14.27040(15) |
| a, b, c (Å) | 26.0868(3) |
| | 9.99412(12) |
| | 90 |
| α, β, γ (°) | 108.3991(7) |
| | 90 |
| Volume ($\text{Å}^3$) | 3530.31(7) |
| Z, Density (calc.) ($mg/m^3$) | 4, 1.307 |
| θ range for data collection (°) | 2.0 to 70 step 0.013 (2θ) |
| Refinement method | Rietveld |
| Data/restraints/parameters | 3477/52/77 |
| Final R indices [I > 2σ(I)] | $R_{wp}$ = 8.61 |
| | $Chi^2$ = 107 ($R_{exp}$ = 0.834) |
| CCDC | 2044269 |

*2.3. Theoretical Methods*

The Gaussian-16 program [22] was used for the calculations (PBE0 functional [23] and def2-TZVP basis set [24,25]). Moreover, the D3 correction [26] has been applied. The energies are BSSE-corrected [27]. The same level has been used for QTAIM and NCI plot analyses by means of the AIMAll package [28]. The MEP surface has been plotted using the 0.001 a.u. isosurface. For the calculations, we have used the experimental coordinates.

## 3. Results and Discussion

*3.1. Description of the Cocrystal*

The unit cell contains 4 Sildenafil and 8 resorcinol molecules. In the crystal structure, Sildenafil forms an interesting hydrogen bond (HB)/π–π/HB assembly where the pyrazolo[3,4-d]pyrimidine rings are π-stacked in an antiparallel binding mode to maximize the dipole⋯dipole interaction. Simultaneously, an N-atom of the pyrazole ring interacts with one OH group of the resorcinol molecule. The assembly is represented in Figure 3, and in the following section, we further analyze the interplay between the H-bond and π-stacking interaction and how the H-bond polarizes the π-system, increasing the dipole moment and concomitantly reinforcing the π–π stacking.

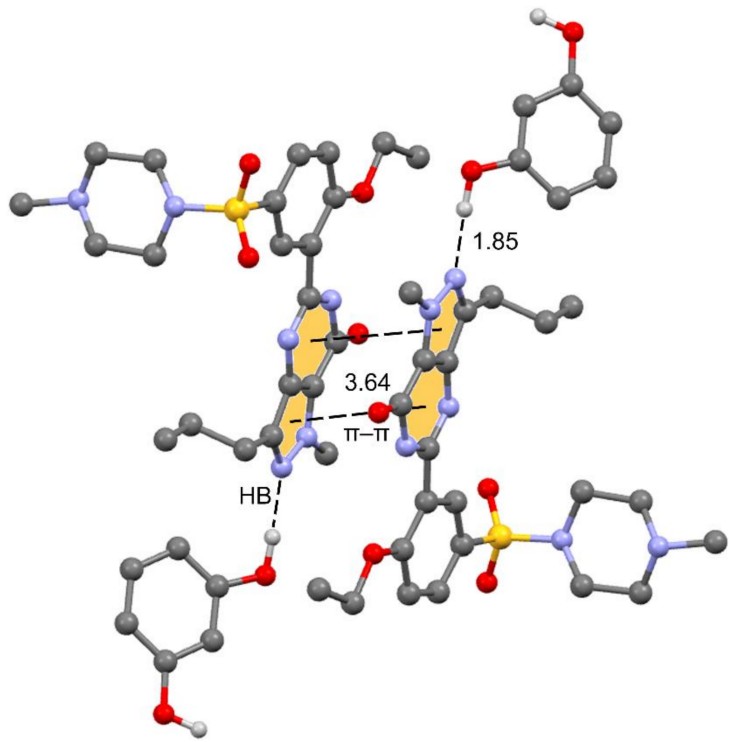

**Figure 3.** Partial view of the cocrystal structure showing the antiparallel π stacking and the Sildenafil···resorcinol H-bonds.

## 3.2. Theoretical Study

First, the MEP surfaces of Sildenafil and the H-bonded complex Sildenafil···resorcinol have been computed and compared. Both surfaces are given in Figure 4, showing that the minimum is located at the O-atom of the sulfonamide group. Both the N- and O-atoms of the pyrazolo[3,4-d]pyrimidine moiety also present large and negative MEP values (−30 kcal/mol). The values over the six- and five-membered rings of the pyrazolo[3,4-d]pyrimidine moiety are small and of opposite sign, thus favoring the antiparallel stacking. In Figure 4b, the MEP surface of the H-bonded complex is represented, showing that the π-system of the pyrazolo[3,4-d]pyrimidine moiety is more polarized than that of isolated Sildenafil. It also evidences that the maximum MEP value corresponds to the H-atom of the phenol group (+45 kcal/mol). The dipole moments are also indicated in Figure 4, confirming that the dipole moment of the H-bonded dimer (8.5 D) is significantly larger than that of Sildenafil (6.4 D), thus anticipating a larger ability to form dipole···dipole interaction in the dimer.

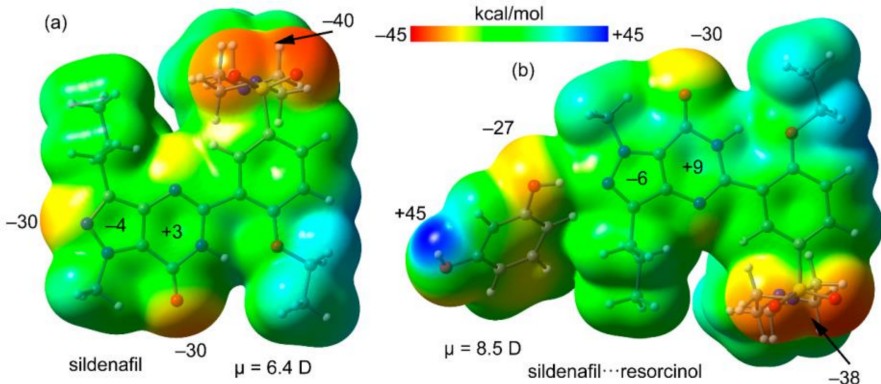

**Figure 4.** Molecular electrostatic potential (MEP) surface of Sildenafil (**a**) and its H-bonded complex with resorcinol (**b**) at the PBE0/def2-TZVP level of theory. Isosurface 0.001 a.u. Energies at selected points in kcal/mol.

In Figures 5 and 6, we show the different reactions used to evaluate the mutual influence of H-bonds and π-stacking interactions that are likely important for the formation of the HB/π–π/HB assembly shown in Figure 3. In Figure 5 (top), the interaction energy of the H-bonded dimer is shown, evidencing a very strong interaction ($\Delta E_1$ = −11.3 kcal/mol). If the π–π stacked dimer is used as a whole entity and starting point for the computation of the formation of the tetrameric assembly (Figure 5, bottom), the resulting formation energy ($\Delta E_2$ = −31.5 kcal/mol) is much greater than twice the H-bonding energy of the dimer ($\Delta E_1$), thus suggesting that the π stacking interaction reinforces the H-bond. Figure 6 (top) shows the formation energy of the π-stacked dimer, which is $\Delta E_3$ = −16.7 kcal/mol. In the case the H-bonded dimer is used to generate the tetrameric assembly (Figure 6, bottom), the formation energy is $\Delta E_4$ = −25.7 kcal/mol, thus suggesting that the H-bond also reinforces the π-stacking. This large reinforcement is attributed to the stronger dipole···dipole interaction, and also additional interactions that are established between the resorcinol and the Sildenafil upon complexation, as further discussed below.

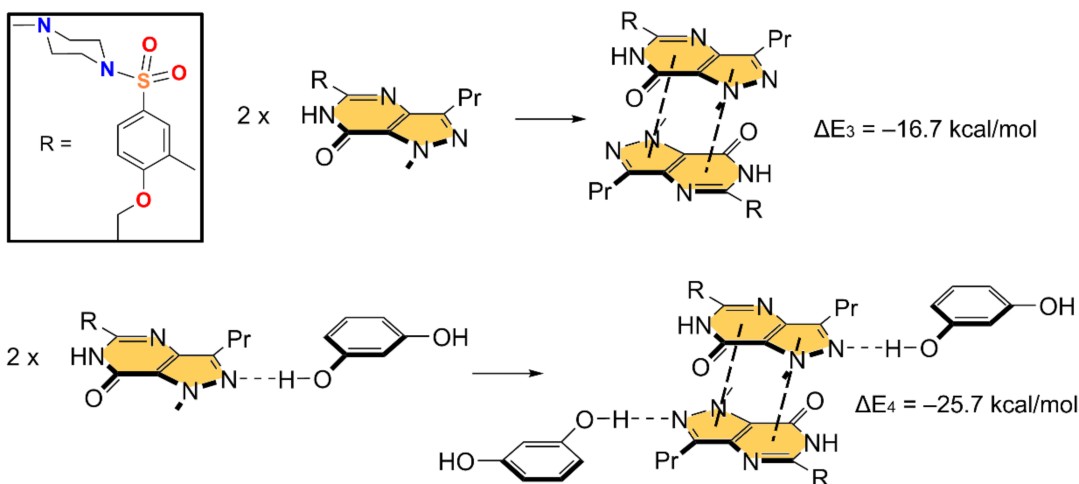

**Figure 5.** Reactions employed to measure the H-bonds in the dimer (**top**) and tetramer (**bottom**).

**Figure 6.** Reactions employed to measure the π-stacking in the dimer (**top**) and tetramer (**bottom**).

We have used the QTAIM and NCI plot analyses to further characterize the H-bonded and π-stacking dimers observed in the structure and to rationalize the large dimerization energies. The results are shown in Figure 7. For the H-bonded dimer, the O–H···N H-bond is characterized by a bond critical point (CP) and bond path connecting the H and O-atoms. The NCI plot shows an intense blue isosurface, which indicates a very strong interaction, in agreement with the strong dimerization energy. Moreover, the QTAIM/NCI plot analysis also evidences the existence of an

additional C–H⋯N H-bond involving the C–H bond adjacent to the interacting OH. The color of the NCI plot surface that characterizes this C–H⋯N bond is green, thus revealing a much weaker interaction (see Figure 7a). We have evaluated the individual contribution of each H-bond in this dimer by using the potential energy density at the bond CP ($V_r$). The energy of the H-bond can be calculated by using the equation proposed by Espinosa et al. (E = ½Vr) [29], and recently used by us to analyze similar interactions [30–35]. By using this formula, the contribution of the O–H⋯O bond is −9.32 kcal/mol ($V_r$ = −0.0297 a.u.) and that of the C–H⋯N bond is −1.31 kcal/mol ($V_r$ = −0.0042 a.u.). The sum of both contributions (−10.63 kcal/mol) is very similar to the dimerization energy, (−11.3 kcal/mol), thus giving reliability to the H-bond energy predictor.

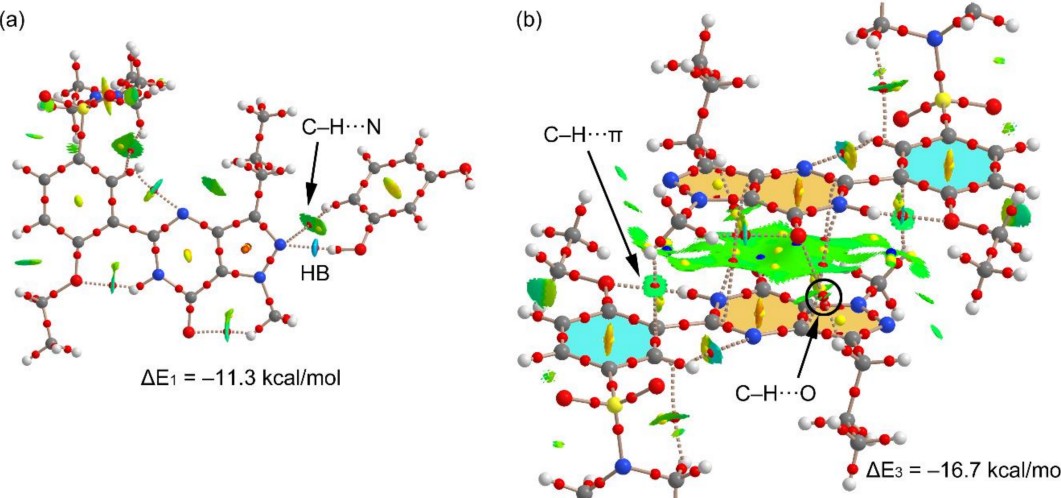

**Figure 7.** Combined QTAIM (bond, ring, and cage critical points (CPs) represented as red, yellow, and blue spheres, respectively) and noncovalent interaction (NCI) plot analysis (−0.04 < sign($λ_2$) ρ < 0.05) of the H-bonded dimer (**a**) and π-stacking dimer (**b**). NCI plot isosurface = 0.5 a.u.

Figure 7b shows the combined QTAIM/NCI plot analysis of the π-stacked dimer. The dimerization energy is very large due to the antiparallel orientation of the dipole and also the existence of secondary interactions, as revealed by the QTAIM distribution of CPs. The π–π stacking interaction is characterized by four CPs and bond paths interconnecting atoms of the pyrazolo[3,4-d]pyrimidine moieties. The QTAIM analysis reveals the existence of two symmetrically equivalent C–H⋯π interactions involving the methyl groups attached to the pyrazole rings and the phenyl rings. Moreover, two equivalent C–H⋯O interactions are also present involving one H-atom of the propyl group and the exocyclic O-atom of the pyrimidine ring.

Finally, to further analyze the secondary interactions in the tetrameric assembly, we have represented the NCI plot in Figure 8. It can be observed that the π–π interaction is characterized by an extended green isosurface and the H-bonds by small and intense blue isosurfaces. The NCI plot also shows several green isosurfaces that are located between the resorcinol ring and the Sildenafil molecule, thus evidencing the existence of additional van der Waals contacts apart from the H-bonds and π-stacking interaction upon the formation of the tetramer (see dashed rectangles in Figure 8). These interactions also contribute to the enhancement of the π-stacking as a consequence of the H-bonds, and vice versa. Therefore, the cooperativity effects evidenced by the reactions gathered in Figures 5 and 6 are also influenced by the long-range interactions between the resorcinol and the Sildenafil molecular fragments not directly H-bonded.

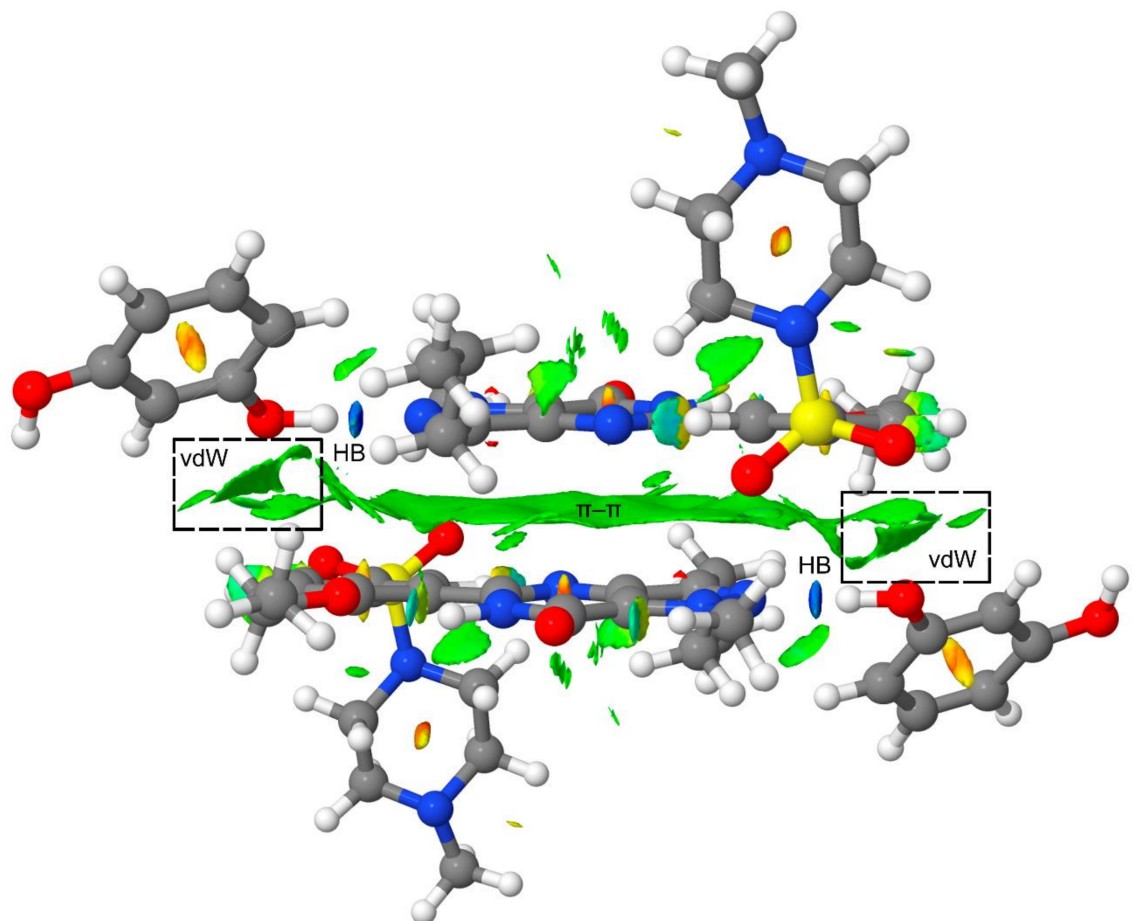

**Figure 8.** NCI plot of the tetrameric assembly at the PBE0-D3/def2-TZVP level of theory. Range: $-0.04 < \mathrm{sign}(\lambda_2)\rho < 0.05$ and RDG isosurface = 0.5 a.u.

## 4. Concluding Remarks

We have determined the crystal structure of a Sildenafil-resorcinol cocrystal from X-ray powder diffraction data by means of direct space methods and analyzed its structural features through DFT calculations. The formation of strong H-bonds and π-stacking interactions has been studied and interpreted in terms of cooperativity effects between both interactions to generate the HB/π-π/HB assembly that is based on the enhanced dipole···dipole interaction. The energetic features of the assembly have been studied using DFT calculations and characterized by the QTAIM and NCI plot computational tools, showing that the π–π stacking interaction is reinforced by the H-bonding and vice versa.

**Supplementary Materials:** The following are available online at http://www.mdpi.com/2073-4352/10/12/1126/s1, Figure S1: prf plot for the crystal structure refinement of Sildenafil–resorcinol cocrystal determined by TALP. The plot shows the experimental powder XRD profile (red marks), the calculated powder XRD profile (black solid line), and the difference profile (blue, lower line). Tick marks indicate peak positions, Figure S2: Comparative plot of the deformed resorcinol molecule: (a) After TALP crystal structure determination; (b) after DFT calculations, Figure S3: (a) Overlapping of Sildenafil-resorcinol cocrystal structure initially determined by TALP (blue) and final structure refined by the Rietveld method (red). The computed root-mean-square distance (RMSD) is 0.0003 Å. (b) Detail of the overlapped resorcinol molecules.

**Author Contributions:** Conceptualization, A.F. and R.P.; methodology, R.B., R.P. and A.F.; software, R.P., A.F. and O.V.; validation, R.P., R.B. and V.K.; formal analysis, R.B. and O.V.; investigation, A.F., R.P. and R.B.; resources, R.P. and A.F.; data curation, R.B.; writing—original draft preparation, A.F. and R.P.; writing—review and editing, A.F. and R.P.; supervision, A.F. and R.P.; project administration, A.F. and R.P.; funding acquisition, A.F. and R.P. All authors have read and agreed to the published version of the manuscript.

**Funding:** This research was funded by MICIU/AEI from Spain, CTQ2017-85821-R, FEDER funds.

**Acknowledgments:** We thank the "centre de tecnologies de la informació" (CTI) at the University of the Balearic Islands for computational facilities.

**Conflicts of Interest:** The authors declare no conflict of interest.

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
