# Peer review of "Sildenafil–Resorcinol Cocrystal: XRPD Structure and DFT Calculations"

_crystals, doi:10.3390/cryst10121126_

Round 1
Reviewer 1 Report
The work is done consistently and conducted very thoroughly. The combination of crystallographic structures and computational approach is merged into a very nice discussion. From the computational point of view, the methodology selected is appropriate as so it is the computational level, including D3 dispersion to assess the interaction described by the authors.
The literature is also cited appropriately and both the introduction and manuscript are concise.
I found the conclusions a bit too concise and I guess it could be extended, but that is totally up to the authors.
I would suggest the acceptance of the manuscript as it stands
Author Response
Thank you for the careful reading of the manuscript and recommending publication.
Comment: I found the conclusions a bit too concise and I guess it could be extended, but that is totally up to the authors.
Reply: We have expanded the conclusions by adding a sentence regarding the cooperativity between both pi-pi and H-bonding interactions.
Reviewer 2 Report
This paper focused on a crystal structure analysis for the sildenafil–resorcinol cocrystal by using X-ray powder diffraction and DFT calculation. The concept and strategy are interesting and also the manuscript is well written and organized as well as experimental design. This research topic and obtained results are expected to attract attention by many researchers in this field. Therefore, the reviewer recommends the publication of this paper in in this journal, Crystals as present form.
Author Response
Thank you for the careful reading of the manuscript and recommending publication.
Reviewer 3 Report
The manuscript ID: 1019829 reports ‘’Sildenafil–resorcinol cocrystal: XRPD structure and DFT calculations’’ In this study, hydrogen bond and pi-pi interactions in a sildenafil-resorcinol cocrystal are investigated by DFT calculations and NCI plot analysis. The emphasis is placed on how the hydrogen bond and pi-pi interaction strengths alter in dimeric and tetrameric assemblies. The work is of potential interest to crystals readers and meets the standards of the journal. I recommend the work for publication after below comments were addressed.
Line 31: References 7 and 8 discuss characteristics of structures based on Sildenafil and do not relate to cardiovascular problems. Provide relevant references here.
Line 38-40: Moreover, for drugs……..ingredient development. REFERENCES??
Line 49: I recommend to use NCI and plot separately instead of NCIplot. The former abbreviation manner is recommended.
Line 57-58: I recommend authors to provide full co-crystal synthesis details in this journal instead of referencing 9. Furthermore, the sentence ‘’………experiments in xylene, (m.p. 152° C).’’ is confusing. Rephrase the sentence.
Line 79: Table 1 ‘’….space group P21/a.’’ 1 must be subscript. P must be italics.
Figure 2: ‘’Tick marks indicate peak positions.’’ I do not see tick marks?
Line 92-94: Cooperativity is a diverse topic. Authors should explain what cooperativity effects they are aiming to study and relevant references must be provided.
Line 97-99: Authors must show the N–H…O(Et) hydrogen bonds in figure 3. This provides the reader with a better understanding of the ahead NCI plot text.
Line 184: ‘’…cocrystal from laboratory…’’ laboratory is confusing. Rephrase the sentence.
Overall, the references must be thoroughly checked to fit the discussion. More importantly, the authors should explain why 1,3-dihydroxybenzene? Were the other dihydroxy derivatives attempted to study? What are the results? The failures must be briefly discussed.
Author Response
Thank you for the careful reading of the manuscript, corrections and suggestions. Our point-by-point responses follow:
Comment: Line 31: References 7 and 8 discuss characteristics of structures
based on Sildenafil and do not relate to cardiovascular problems.
Provide relevant references here.
Reply: The references 7 and 8 are intended for "acetylsalicylic acid and also in the form of salicylate salt" not for "cardiovascular problems. So, in order to avoid the confusion references 7 and 8 have been relocated in other lines of the same paragraph.
Comment: Line 38-40: Moreover, for drugs……..ingredient development. REFERENCES??
Reply: DONE! thank you
Comment: Line 57-58: I recommend authors to provide full co-crystal synthesis
details in this journal instead of referencing 9. Furthermore, the sentence ‘’.experiments in xylene, (m.p. 152° C).’’ is confusing. Rephrase the sentence.
Reply: DONE
Comment: Line 79: Table 1 ‘’….space group P21/a.’’ 1 must be subscript. P must
be italics.
Reply: DONE! thank you
Comment: Figure 2: ‘’Tick marks indicate peak positions.’’ I do not see tick marks?
Reply: They are in blue, a short text has been added in this sense.
Comment: Line 184: ‘’…cocrystal from laboratory…’’ laboratory is confusing.
Rephrase the sentence.
Reply: "laboratory" has been removed.
Comment: Overall, the references must be thoroughly checked to fit the
discussion. More importantly, the authors should explain why
1,3-dihydroxybenzene? Were the other dihydroxy derivatives attempted
to study? What are the results? The failures must be briefly
discussed.
Reply: All this is explained in our previous work (reference 9). Our current manuscript only discusses the crystal structure of one of the cocrystals tested in the work from reference 9. Honestly, we think that a further discussion on that would not be relevant in the context of the results described in our current manuscript, so we kindly ask the reviewer to keep this section as it is.
Reviewer 4 Report
The Authors report an interesting study concerning the X-Ray Powder Diffraction (XRPD) crystal structure of a new sildenafil cocrystal, where resorcinol has been used as the coformer. The crystal structure has been solved by means of direct space methods used in combination with density functional theory (DFT) calculations. Interestingly, in the structure the sildenafil and resorcinol molecules form cooperative hydrogen bond (HB) and π-stacking interactions that have been analyzed using DFT calculations, molecular electrostatic potential (MEP) surface and noncovalent interaction plot (NCIPlot). In particular, the formation of O–H···N H-bonds between resorcinol and sildenafil increases the dipole moment and enhances the antiparallel π-stacking interaction.
In my opinion, the manuscript is well presented and discussed. The results described in the manuscript strongly support the conclusions of the study. In my opinion the manuscript deserves to be considered for publications on Crystals.
Minor comments:
- Please check and insert the final full stop in captions of Figures 1, 4 and 7.
- Maybe, sildenafil should be written with the S in capital letter (for example in caption of Figure 1 but also through the main text).
- References section: the titles should be written in a homogeneous form. Please check the use of capital letter, and the use of full stop or comma.
Author Response
Thank you for the careful reading of the manuscript, corrections and suggestions. Our point-by-point responses follow:
Comment: Please check and insert the final full stop in captions of Figures 1, 4 and 7.
Reply: Done
Comment: Maybe, sildenafil should be written with the S in capital letter (for example in caption of Figure 1 but also through the main text).
Reply: Done
Comment: References section: the titles should be written in a homogeneous form. Please check the use of capital letter, and the use of full stop or comma.
Reply: Done